

# Rhamnolipids bio-production and miscellaneous applications towards green technologies: a literature review

Sanaa S.A. Kabeil[1,*], Amira M.G. Darwish[2,3,*], Soad A. Abdelgalil[4], Abdelaal Shamseldin[5], Abdallah Salah[1], Heba A.I.M. Taha[6], Shimaa Ismael Bashir[7], Elsayed E. Hafez[7] and Hesham Ali El-Enshasy[8,9,10]

[1] Protein Research Department, Genetic Engineering and Biotechnology Research Institute (GEBRI), City of Scientific Research and Technological Applications (SRTA-City), Alexandria, Borg El Arab, Egypt
[2] Food Industry Technology Program, Faculty of Industrial and Energy Technology, Borg Al Arab Technological University (BATU), Alexandria, Borg El Arab, Egypt
[3] Food Technology Department, Arid Lands Cultivation Research Institute, City of Scientific Research and Technological Applications (SRTA-City), Alexandria, Borg El Arab, Egypt
[4] Bioprocess Development Department, Genetic Engineering and Biotechnology Research Institute (GEBRI), City of Scientific Research and Technological Applications, (SRTA-City), Alexandria, Borg El Arab, Egypt
[5] Enviromental Biotechnology Department, Genetic Engineering and Biotechnology Research Institute (GEBRI), City of Scientific Research and Technological Applications, (SRTA-City), Alexandria, Borg El Arab, Egypt
[6] Department of Nutrition, High Institute of Public Health, Alexandria University, Alexandria, Egypt
[7] Department of Plant Protection and Biomolecular Diagnosis, Arid Land Cultivation Research Institute, City of Scientific Research and Technological Applications (SRTA-City), Alexandria, Borg El Arab, Egypt
[8] City of Scientific Research and Technology Application (SRTA-City), Alexandria, Egypt
[9] Universiti Teknologi Malaysia (UTM), Innovation Centre in Agritechnology for Advanced Bioprocessing, Johor, Malaysia
[10] Faculty of Chemical and Energy Engineering, Universiti Teknologi Malaysia, Johor, Malaysia
* These authors contributed equally to this work.

Corresponding authors
Amira M.G. Darwish,
amiragdarwish@yahoo.com
Hesham Ali El-Enshasy,
henshasy@ibd.utm.my

## ABSTRACT

Growing attention towards rhamnolipids (RLs) biosurfactants with antibacterial, anti-fungal, antivirus and antitumor potentials encourage future research in biotechnology and biomedicine fields. Economic production from waste materials, biodegradability and low toxicity makes RLs perform as green molecules that serve in sustainability and green technologies. This review aims to focus on bioproduction, detection and applications of rhamnolipids in pharmaceuticals, soil bioremediation, agriculture and food industries in addition to future perspectives. This will help to shed light on and update the existing knowledge of feasible and sustainable biosurfactant production depending on the fermentation processes.

## INTRODUCTION

A subclass of glycolipids known as rhamnolipids (RL) is made up of mono-, di- or oligosaccharides and lipids in which various sugars (such as glucose, mannose, galactose,

glucuronic acid, or rhamnose) Mono-rhamnolipids contain one rhamnose moiety and di-rhamnolipids contain two (*Behrens et al., 2016*), connect to generate saturated or unsaturated fatty acids, hydroxylated fatty acids, or fatty alcohols (*Araujo et al., 2022*). Rhamnolipid production by a variety of bacteria, primarily *P. aeruginosa*, in liquid culture under growth-limiting circumstances using hydrocarbons and other hydrophobic carbon sources is well documented (*Müller et al., 2010*). *P. aeruginosa* rhamnolipids exhibit good biodegradability, low toxicity, and environmental friendliness (*Marsudi, Unno & Hori, 2008*), while having physicochemical characteristics similar to those of artificial surfactants.

Different types of surface-active chemicals known as "biosurfactants" are created by microbes. The vast majority of surface-active substances are chemically produced. Consideration of biological surfactants as alternatives to chemically produced chemicals is a result of growing environmental concern. Due to their low toxicity and ability to degrade, biosurfactants have many advantages over synthetic surfactants in terms of environmental acceptance. Using cutting-edge surfactants like those made by microorganisms can help to reduce greenhouse gas emissions (*Marchant & Banat, 2012*).

Producers of microbial biosurfactants offer products made up of a diverse range of congeners based on a fundamental structure. The different topologies have an impact on a variety of molecular characteristics, including water solubility and micelle structure. Commercial applications require a certain quality of the utilized surfactants (*De Almeida et al., 2016*). Microorganisms produce surface-active compounds to survive under low moisture and inaccessible substrate circumstances. Biosurfactant production requires the presence of miscible hydrophilic and oily/hydrocarbon-type carbon sources in the culture medium. Biosurfactants rhamnolipids including rhamnose and β-hydroxy decanoic acid are discussed in terms of their microbial producers, physiological role, biosynthesis, genetics, microbial overproduction, physicochemical features, and possible uses in a variety of sectors (*Nitschke, Costa & Contiero, 2005*).

The principal types of high-mass surfactants are referred to as bioemulsions and comprise polymeric and particulate surfactants, as well as glycolipids, lipopeptides, phospholipids, flavolipids, and corynomycolic acids (*Salihu, Abdulkadir & Almustapha, 2009*). These high molecular weight biosurfactants display significant substrate selectivity and are extremely effective emulsifiers that function at low doses (*Banat et al., 2010*; *Salihu, Abdulkadir & Almustapha, 2009*).

The hydrophobic portion of most biosurfactants is made up of long-chain fatty acids or fatty acid derivatives, whereas the hydrophilic component can be made up of a carbohydrate, amino acid, phosphate, or cyclic peptide (*Nitschke & Costa, 2007*; *Muthusamy et al., 2008*).

Evonik was granted a patent in April 2017 regarding the production of rhamnolipids from butane by recombinant *Pseudomonas putida* strain 20. Despite the advances in technology, there is still the need to further decrease production costs to achieve a competitive cost-effective process. The technical challenges works to offering alternative strategies producing mono-RLs from sucrose with genetically engineered GRAS yeast strains. The production of recombinant strains demonstrated the potential of a yeast based heterologous RLs and rhamnose production (*Bahia et al., 2018*). Ever since, the recombinant *P. putida* was used

applying various carbon sources such as waste vegetable oil, peanut powder, and glycerol for efficient synthesis of rhamnolipids (*Pang et al., 2024*).

## Search methodology

This literature review goal is to focus on bioproduction, the potential applications of rhamnolipids in pharmaceuticals, soil bioremediation, agriculture and food industries in addition to future perspectives, especially with their positive economic and environmental impact. To achieve our goal of ensuring comprehensive and unbiased coverage of the literature, we targeted highly peer-reviewed, especially CC-BY-4.0 licensed articles published in English, starting from the year 2000 (except for a few basic articles), with no geographical location limitation including: case studies, research articles, review articles and market analyses employing search engines, mainly Google and Firefox. In order to be precise in our search, some search terms were used such as Rhamnolipids, Bioproduction, Biosurfactants, Bioemulsifier, Functional properties, Biotechnology and Green economy. For further precision, the exclusion criteria included factors or characteristics related to genetics. These factors cannot be well covered within the current data which may confuse the readers and affect the targeted outcomes. Our efforts are primarily motivated by a desire to learn more about different bioproduction, and the potential applications of rhamnolipids toward green economy.

## Rationale and intended audience

Biosurfactants are safe, multi-functional, emulsification, micelle production, and oil displacement. From environmental perspective, biosurfactants are biodegradable, environmentally compatible, and non-toxic. The sustainability of science thus requires well-behaved alternates that best suit the demand. Integration with green economy recommends valorization of agro-food wastes as suitable feedstock for added value sustainable biosurfactant production to mitigate global carbon footprint and contribute to the United Nations SDGs goal. The biosurfactants' growing market size was valued for over USD 8 billion in 2022 and is set to depict around 5% CAGR from 2023 to 2032. Thus, this review may attract the attention of a wide array of audience in fields related to biosurfactants production, industrial applications, environmental sciences, microbiology and marketing.

## RLS PRODUCING MICROORGANISMS

The majority of RLs are isolated from *Pseudomonas* species, as *P. aeruginosa* is the main producer. Two *Acinetobacter calcoaceticus* isolates from the family Moraxellaceae, order Pseudomonadales were discovered to synthesize RL (*Rooney et al., 2009*). However, the majority of additional cases fall within the phylum (Gammaproteobacteria) of various orders; *e.g.*, *Pseudoxanthomonas* sp. (order: Xanthomonadales; family: Xanthomonadaceae) (*Nayak, Vijaykumar & Karegoudar, 2009*), and *Pantoea* sp. (order: Enterobacteriales; family: Enterobacteriaceae) as well as *Enterobacter* sp. (*Andrä et al., 2006*). Several Betaproteobacteria called *Burkholderia* sp. produce RL (*Dubeau et al., 2009*). *Myxococcus* sp. (Deltaproteobacteria) strain can produce unusual rhamnose-containing glycolipids
(rhamnosides) called myxotyrosides. These lipids have a tyrosine-derived core structure that is glycosylated with rhamnose and acylated with uncommon fatty acids like (Z)-15-methyl-2hexadenoic (*Rahman & Gakpe, 2008*). Other phyla such as Salmonid-specific *Renibacterium* (*Christova et al., 2004*), the bacterium *Cellulomonas cellulans* (*Arino, Marchal & Vandecasteele, 1998*), *Nocardioides* sp. (*Vasileva-Tonkova & Gesheva, 2005*), have claimed to produce RLs (*Almuhayawi et al., 2021*). *Tetragenococcus koreensis*, a member of the phylum Firmicutes is a generator of RL (*Lee et al., 2005*). This emphasizes the necessity of broad-spectrum screening programs for new RLs-producers, which should cover other orders of Gammaproteobacteria or possibly other classes (*Rooney et al., 2009*), This is advantageous from a biotechnological perspective because it might lead to the discovery of producers that are less pathogenic than *P. aeruginosa* strains and are better candidates for the synthesis of RLs in an industrially safe manner. Table 1 summarizes the diversity of some RLs-producing bacteria (*Abdel-Mawgoud, Lépine & Déziel, 2010*). Recently, bioactivity screening of *Paraburkholderia* spp. isolates discovered their production of a set of rhamnolipid surfactants with a natural methyl ester modification. These rhamnolipid methyl esters exhibited enhanced antimicrobial activity against pyrophilous *Pyronema* fungi and *Amycolatopsis* bacteria, and showed enhanced surfactant properties and facilitated bacterial motility on agar surfaces, compared to the rhamnolipids made by *Pseudomonas* spp (*Liu et al., 2024*). *Burkholderia thailandensis* is a potent non-pathogenic rhamnolipid producer. Despite of its numerous advantages, the research still requires additional efforts for investigating regulatory mechanisms and substrate utilization. Natural competence of *B. thailandensis* for genetic manipulations can be used for creating strains with (i) enhanced pathways, (ii) better expression of important enzymes, (iii) efficient nutrient bioconversion and (iv) rapid product export outside the cell. Recently, thermophilic *Geobacillus stearothermophilus* was reported as a promising rhamnolipid biosurfactant producer that utilizes many organic wastes. The produced biosurfactant could be applied as a promising emulsifier, antimicrobial, antioxidant, and plant growth promoter (*Albasri et al., 2024*). Rhamnolipids production from such well-established microbial cell factories could replace the conventional surfactants, lower the carbon footprint, and make the rhamnolipid applications environment friendly (*Kumar et al., 2023*).

## RHAMNOLIPID PRODUCTION FACTORS

The final quantity and quality of the produced RLs depends on several internal and external factors as illustrated in Fig. 1. These factors either related to the medium such as components, pH, temperature, agitation, aeration, dissolved oxygen, fermentation time. To ensure the maximum yield paired with lower cost; medium optimization strategies are required. To achieve this target, researchers use statistical optimization strategies based on response surface methodology (RSM) that guarantee the optimum condition with feasible production cost (*Begum, Saha & Mandal, 2023*).
**Table 1 The diversity of RLs-producing bacteria (after *Abdel-Mawgoud, Lépine & Déziel, 2010*).**

| Bacterial sp. | Culture medium | RL composition |
|---|---|---|
| *B. glumae* | Not mentioned | Rha-Rha-$C_{14}$-$C_{14}$, $C_{12}$-$C_{14}$, and $C_{14}$-$C_{16}$ (beside their isomers) |
| *Cellulomonas cellulans* | Mineral salts+yeast extract+3% glycerol or 2% *n*-hexadecane | Rhamnose-containing glycolipid (glucorhamno-ribo-lipid) |
| *Acinetobacter calcoaceticus* | MSM+10% glycerol | Mono- and di-RL with $C_{10}$-$C_{10}$ |
| *Pantoea stewartii* | MSM+10% glycerol | Mono- and di-RL with $C_{10}$-$C_{10}$ |
| *Pantoea sp.* | MSM+2% *n*-paraffin or kerosene | RL[a] (congeners unidentified) |
| *B. plantarii* | Nutrient broth (NB) +4% glycero Mineral salts+yeast ext+soy bean oill | Rha-Rha-$C_{14}$, Rha-Rha-$C_{14}$-$C_{14}$, and Rha-Rha-$C_{14}$-$C_{14}$-$C_{14}$ |
| *P. clemancea* | MSM+3% glycerol+high phosphate+no trace elements | Mono- and di-RL with C10-C10 |
| *Enterobacter hormaechei* | MSM+10% glycerol | Mono- and di-RL with $C_{10}$-$C_{10}$ |
| *Nocardioides sp.* | MSM+2% *n*-paraffin | RL[a] (congeners unidentified) |
| *Renibacterium salmoninarum* | Mineral salts medium (MSM) +2% *n*-hexadecane or *n*-paraffin | Mono- and di-RL |
| *B. pseudomallei* | Vogel–Bonner medium, glycerol medium | Di-RL congeners with $C_{12}$-$C_{12}$, $C_{12}$-$C_{14}$, $C_{14}$-$C_{14}$, $C_{14}$-$C_{16}$ and $C_{16}$-$C_{16}$ |
| *P. stutzeri* | MSM+glucose 10% | RL[a] (congeners unidentified) |
| *Myxococcus sp.* | Peptone medium+0.2% starch+0.2% glucose | Myxotyrosides A and B which are rhamno-amino-lipids |
| *Enterobacter asburiae* | MSM+10% glycerol | Mono- and di-RL with $C_{10}$-$C_{10}$ |
| *P. fluorescens* | Bushnell–Haas Broth+0.1 mg yeast, 0.1 X NB+2,000 IU penicillin | RL[a] (congeners unidentified) |
| *P. putida* | Sugar beet molasses at 5% w/v | RL[a] (congeners unidentified) |
| *P. putida* | Mineral salt agar+2% hexadecane | RL[a] (congeners unidentified) |
| *P. fluorescens* | Nutrient broth | RL[a] (congeners unidentified) |
| *P. alcaligenes* | Nutrient agar (NA), mineral medium+0.5% (v/v) of palm oil | Mono- and di-RL with $C_8$-$C_{10}$, $C_{10}$-$C_{10}$, and $C_{10}$-$C_{12}$ |
| *P. aeruginosa* | MSM+soluble or insoluble carbon sources | |
| *P. cepacia* | Nutrient broth | RL[a] (congeners unidentified) |
| *P. stutzeri* | MSM+crude oil 1% | RL[a] (congeners unidentified) |
| *P. stutzeri* | Nutrient broth | RL[a] (congeners unidentified) |
| *P. collierea* | MSM+3% glycerol+high phosphate+no trace elements | Mono- and di-RL with C10-C10 |
| *P. fluorescens* | MSM+2% n-paraffin or kerosene | RL[a] (congeners unidentified) |
| *P. luteola* | Sugar beet molasses at 5% w/v | RL[a] (congeners unidentified) |
| *P. putida* | Mineral salts+yeast extract+glucose+corn oil | RL, the exact structures were not determined |
| *P. putida* | Mineral salts medium+phenanthrene | Suggested to be RLa |
| *P. teessidea* | MSM+3% glycerol+high phosphate+no trace elements | Mono- and di-RL with C10-C10 |

## External environmental factors

PH is always required for the growth of microbial and the fermentation of biosurfactants. In *Pseudomonas aeruginosa,* surface tension measurements differs due to the carboxylic acid moiety (pKa 5.5 for rhamnolipid aggregates), solution adjusted to pH 4.0 or 8.0 create solutions of the protonated nonionic or deprotonated anionic mono-rhamnolipids,

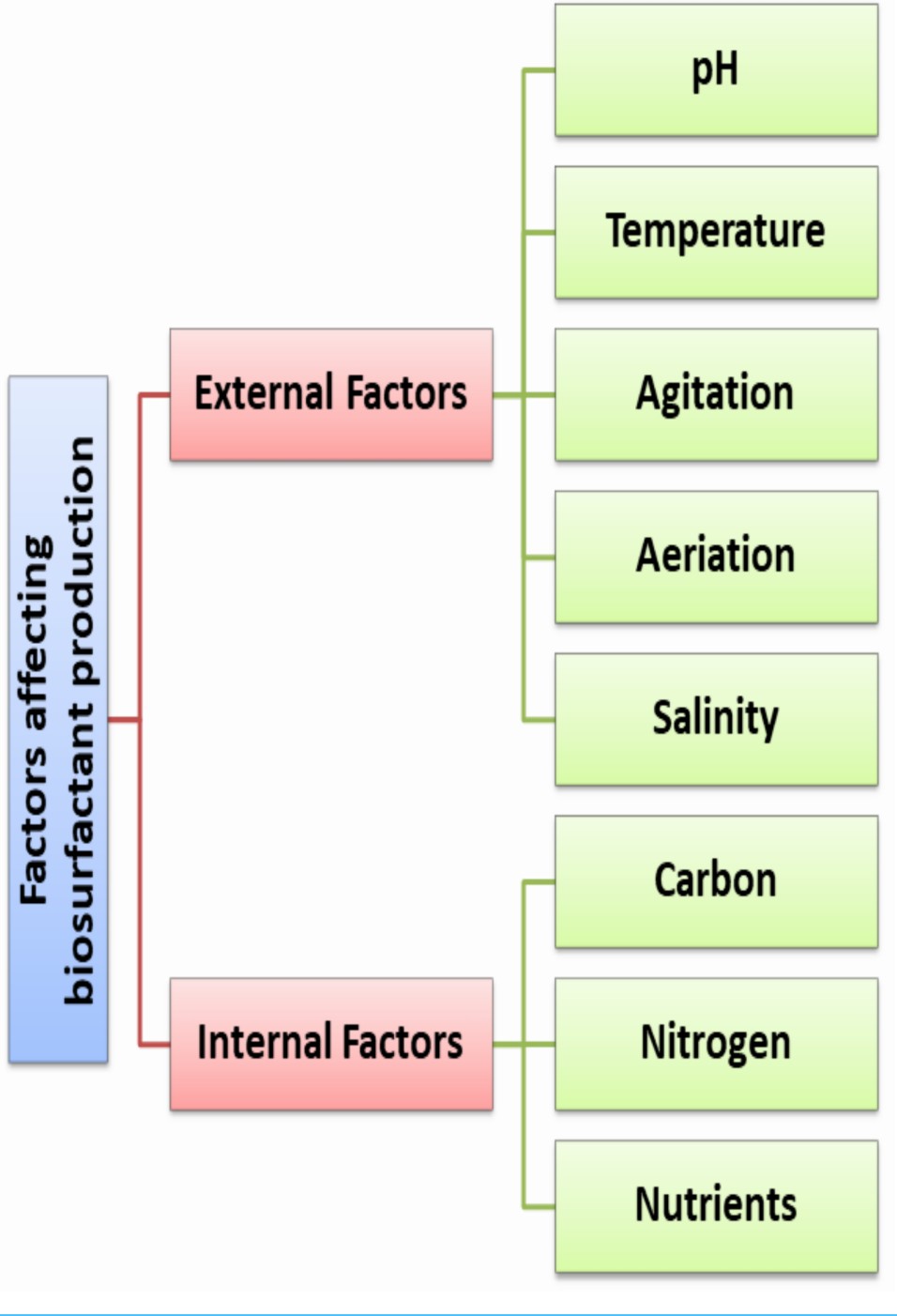

Figure 1   Factors affecting biosurfactant production.

respectively (*Palos Pacheco et al., 2017*; *Cerqueira Dos Santos et al., 2024*). In *C. antarctica* biosurfactants production, the best production was obtained at pH ranged from four to eight (*Begum, Saha & Mandal, 2023*).

Temperature is one of the most important environmental factors in biosurfactant production. The growth of microorganisms is associated with a particular temperature range. Thus, the fermentation of carbohydrates needs an optimum well-maintained temperature due to its impact on the metabolic process and physical properties of the fermentation broth. In case the temperature exceeds the optimum level, denaturation of enzymes and other proteins may occur (*Begum, Saha & Mandal, 2023*). The optimal temperature for the synthesis of rhamnolipids by *P. aeruginosa* 44T1 on glucose was determined to be 37 °C (*Robert et al., 1989*). Others claimed that when *Pseudomonas* sp. DSM 2874 employed n-alkanes as a substrate, it was feasible to control the rhamnolipid composition of the surfactant consisting of four compounds by temperature (*Syldatk et al., 1985*).

As temperature has a broad effect on bacterial metabolism and regulation, different hypothesis may provide an explanation for this behavior. RNA-thermometers are generally reported to display a comparably high degree of "leakiness" and inefficiency when inhibiting translation, which could result in differences in rhamnolipid production rates. In many pathogenic bacteria, expression of virulence genes is induced at 37 °C, and the underlying mechanisms are often based on RNA-thermometers. The presence of an RNA-thermometer in the rhlAB operon coding for the rhamnosyltransferase complex required for rhamnolipid biosynthesis (*Noll et al., 2019*).

Agitation and aeration are very important factors in the fermentation process as (1) they affect the activities of heat, nutrients and oxygen distribution in the medium, (2) they affect medium viscosity, (3) they provide the oxygen necessary for cell growth and fermentation process, (4) they reduce the exhaust gases generated during the process (*Begum, Saha & Mandal, 2023*).

Salinity of the halotolerant strain and many marine and petroleum-contaminated soils that have a saline environment is crucial. The effect of salinity on *Pseudomonas aeruginosa* performance in the production of biosurfactants was measured by the surface tension of media. The reduction of surface tension in media containing *Pseudomonas aeruginosa* should be maintained over 1.4% NaCl concentration for the best quality (32.1 mN/m) (*Ahmadi et al., 2021*).

### Internal factors

Carbon source can affect the produced biosurfactant amount, composition and quality. Various carbon sources could be applied for the production of biosurfactants. Long chain fatty acids such as in soybean oil are applied as carbon source in the rhamnolipids fermentations. Their de-foaming properties can weaken liquid films by competing with foaming metabolites and destabilize the formed bubbles, thus, suppressing the rhamnolipids fermentation broth foaming behavior (*Gong et al., 2021*).

Nitrogen source is an important factor that affects the quality and production rate of biosurfactants including; ammonium chloride, urea and ammonium sulfate. The nitrogen addition showed to effect on cell growth and reduced the surface tension. Furthermore, the nitrogen source in the environment is very important for metabolism especially cellular synthesis and bacterial functions. For production of biosurfactant by *Pseudomonas*

*fluorescens*, olive oil and ammonium nitrate were used, as the best carbon and nitrogen sources, respectively.

Micro- and macronutrients are essential factors affecting RLs production. Broth medium supplemented with metal ions such as $MgCO_3$ and $Fe_2+$ can enhance microbial rhamnolipids production Cassava wastewater represents a low-cost culture medium rich in macro-nutrients (proteins, starch, sugars) and micro-nutrients (iron, magnesium) (*Begum, Saha & Mandal, 2023*; *De Oliveira Schmidt et al., 2023*).

In conclusion, for rhamnolipid production process, all of these issues should be taken into account, demonstrating the need for a multidisciplinary strategy integrating several disciplines and technologies.

## RHAMNOLIPIDS PROPERTIES AS BIOSURFACTANTS

Biosurfactants called rhamnolipids have unique physicochemical characteristics. Rhamnolipids' properties are dependent on their homologous composition and distribution, which are determined by the bacterial strain, the growth circumstances, and the makeup of the medium (*Nitschke, Costa & Contiero, 2005*). Because rhamnolipids' surface activity, solubility, and uses are pH-dependent, it is crucial to take their pKa into account. Rhamnolipids are weak acids because of the terminal carboxylic group; for instance, the potentiometric titration of a mono-rhamnolipid combination in water revealed that the pKa was 4.28 0.16 and 5.50 0.06 for concentrations below and over the rhamnolipids CMC, respectively (*Lebrón-Paler et al., 2006*). When pH is more than 4.0, rhamnolipid molecules behave as anions. These substances have strong solubility in alkaline aqueous solutions as well as in methanol, chloroform, and ethyl ether. However, rhamnolipids are best known for their surface-active properties (*Nitschke, Costa & Contiero, 2005*).

Excellent detergent, emulsifying, foaming, and/or dispersing characteristics are produced by RLs because they can decrease surface and interfacial tensions. For instance, RLs decrease the surface tension of water from 72 to 30 mN/m and the interfacial tension of water/kerosene systems from 43 to 1 mN/m (*Pinzon-Gamez, 2009*). They have low critical micelle concentration (CMC) (5–65 mg/L), high affinity for hydrophobic chemical compounds, and high average emulsifying activity of 10.4 to 15.5 U m/L filtrate (*Rahman & Gakpe, 2008*). The hydrophilic/lipophilic balance (HLB) and the critical micelle concentration (CMC) are two characteristics that are frequently used to describe a surfactant. The HLB of a rhamnolipid generated by *P. aeruginosa* UG2 was calculated to be 17.0 using a correlation between the HLB and CMC for sodium carboxylic acids and 24.1 utilizing group contributions (*Noordman & Janssen, 2002*).

## RHAMNOLIPID TYPES

When *P. aeruginosa* grows on water-immiscible surfaces, two primary forms of rhamnolipids are commonly secreted (*Lang & Trowitzsch-Kienast, 2002*). The most common mono- and di-rhamnolipids, namely α-L-rhamnopyranosyl- 3-hydroxydecanoyl-3-hydroxydecanoate (Rha-C10-C10) and α-L-rhamnopyranosyl-(1–2), contain β–hydroxy

decanoic acid residues (*Müller et al., 2010*). Rhamnopyranosyl-3-hydroxydecanoyl-3-hydroxydecanoate is a chemical compound (Rha-C10-C10) that was demonstrated to have mono-rhamnolipid as the main constituent (*Pornsunthorntawee, Chavadej & Rujiravanit, 2009*). Additionally, two other rhamnolipids with only one fatty acid moiety rhamnolipids are discovered in lower amounts. *Pseudomonas* sp. DSM 2874 resting cell cultures were used to describe these rhamnolipids initially (*Syldatk et al., 1985*). Taxonomic analysis of *rhl* genes from strains of *P. aeruginosa, B. thailandensis* and *B. pseudomallei*, revealed that they belong to the same group in the phylogenetic tree. The presence of an *rhl*A ortholog in *P. fluorescens* SWB25, suggesting that *rhl*A might be involved in other pathways within the organism apart from rhamnolipid production (*Irorere et al., 2017*). Table S1 illustrates the chemical structure of some identified rhamnolipids (*Abdel-Mawgoud, Lépine & Déziel, 2010*).

More than 27 rhamnolipids have been described; they mostly differ in the length of the chain and the level of fatty acid molecule saturation. Rhamnolipids A and B are two additional rhamnolipid species that resemble mono- and di-rhamnolipids but differ in having an extra decanoyl moiety at the terminal rhamnoses (*Leitermann et al., 2010*). It has been suggested that the age of the culture and the chosen bacterial strain are responsible for the differences in the kind, and percentage of rhamnolipid molecules in the ejected biosurfactant (*Pornsunthorntawee, Chavadej & Rujiravanit, 2009*), substrate contents and particular culture circumstances (*Nitschke, Costa & Contiero, 2005*) with the fermenting plan. Conditions and extraction procedures of rhamnolipids produced by *P. aeruginosa* are exhibited in Table S2 (*Eslami, Hajfarajollah & Bazsefidpa, 2020*).

Table S3 demonstrates several biosurfactant types and the microorganisms responsible for them. Take note that bacteria are the main group of organisms responsible for creating surfactants, but certain yeast species are also present (*Nitschke & Costa, 2007*; *Rahman & Gakpe, 2008*).

## SCREENING METHODS OF RLS

Research on strain selection, metabolic engineering, and process development for better rhamnolipid synthesis is dependent on the precise measurement and quantification of rhamnolipids. Currently, analytical, indirect, or colorimetric techniques are used to analyze rhamnolipids (*Pinzon-Gamez, 2009*).

### Analytical methods

Chromatographic techniques such as thin-layer chromatography (TLC) and high-performance liquid chromatography coupled with mass spectrometry (HPLC-MS) are available analytical methods for determining rhamnolipids (*Heyd et al., 2008*). TLC is generally used to analyze culture broth extracts of rhamnolipid composition. Based on either the number of sugars in the molecule (mono-rhamnolipids or di-rhamnolipids) or the length of the fatty acid alkyl chain, normal or reversed phase TLC has produced effective separation of rhamnolipid combinations (*Long et al., 2013*). Rhamnolipids are not accurately quantified by TLC, and considerable sample amounts are required (in comparison to those needed for other tests). However, HPLC-MS not only distinguishes
between various rhamnolipid species but also offers precise structural details and quantification of the rhamnolipids present in the sample. For optimal results, reversed-phase silica columns and negative electrospray ionization are typically utilized in this approach (*Déziel et al., 2000*; *Pinzon-Gamez, 2009*). Although this technique is quite accurate at analyzing rhamnolipids, the equipment is very expensive, and prior purification procedures are necessary to get rid of salts and other impurities that can interfere with the ionization of rhamnolipids in the MS. Losses of analyte due to manipulation during the purification process might also affect the quantitation (*Heyd et al., 2008*; *Pinzon-Gamez, 2009*). Rhamnolipids in aqueous solutions can be identified and measured using an alternate technique such as ATR-FTIR (*Leitermann et al., 2010*). Although this method similarly depends on the availability of expensive equipment, it is suitable for the quick and repeatable examination of rhamnolipids. The procedure still needs to be standardized and its accuracy needs to be increased.

## Indirect methods
### Surface tension indirect methods for rhamnolipid analysis

Tensiometers and goniometers, and methods like the pendant drop and ring method, are common tools for measuring surface and interfacial tension (*De Lima et al., 2009*; *Rooney et al., 2009*). The surface tension and total rhamnolipid concentration are often correlated using these procedures using a calibration curve with pure rhamnolipids as the standard. The main drawbacks of these techniques include the equipment's accessibility, interference from other surface-active substances, and the variable surface tension with each rhamnolipid species (*Pinzon-Gamez, 2009*).

Recent simple method was developed for quantifying biosurfactants by their ability, on paper, to reduce surface tension of aqueous solutions, causing droplet dispersion on an oiled surface in correlation with biosurfactant content. This method was validated with rhamnolipids, surfactin, sophorolipids, and ananatoside B; all are anionic microbial surfactants. Linear ranges for quantification in aqueous solutions for all tested biosurfactants were between 10 and 500 µM. The method showed time-dependent biosurfactant accumulation in cultures of *Pseudomonas aeruginosa* strains PA14 and PAO1, and *Burkholderia thailandensis* E264. This simple assay provided an opportunity to quantify biosurfactant contents of aqueous solutions, for a diversity of surfactants, by means readily available in any laboratory (*Sass et al., 2023*).

### Hemolytic activity

There are two methods for performing tests for hemolytic activity. First, on blood agar plates differential medium containing mammalian blood (often sheep), that shows hemolytic activity of the red blood cells around the bacterial colonies. This reaction is not exclusive to rhamnolipids because it also occurs with lytic enzymes like protease and lysozyme (*Heyd et al., 2008*). Second, quantitative tests for hemolytic activity which are performed using erythrocyte suspensions and measuring the released hemoglobin absorbance at 540 nm. One hemolytic unit is defined as the least amount of hemolysin that completely lyses the blood cells. The drawbacks are the same as those previously mentioned (*Heyd et al., 2008*).

### Agar plate overlaid with hydrocarbons

The appropriate temperature is used to incubate pure isolates for one week after streaking on oil-coated agar plates. Colonies with an emulsified halo around them are identified as BS producers. This is an effective method as the presence of an emulsified halo around the culture serves as a clear indication of the presence of BS (*Satpute et al., 2010*).

### Cell surface hydrophobicity technique

Cell surface hydrophobicity and biosurfactant synthesis are directly correlated. The cells are separated by centrifugation (12,000 g for 30 min/4 °C), twice rinsed in 50 mM phosphate buffer (pH 7.0), and then re-suspended in the same buffer with an A600 of 0.5 (*Satpute et al., 2010*). Hydrocarbons (0.5 mL) and cell suspensions (0.3 mL) are combined, vortexed for three minutes, and then allowed to settle for ten minutes to allow the hydrocarbon phase to rise entirely. To test A600, the aqueous phase is taken out and put in a one mL cuvette. The hydrophobicity (H%) of the cell surface is measured by the drop in the aqueous phase's A600, which is determined as follows;

$$H\% = [(A0 - A)]/A0 \times 100$$

where A0 and A before and after mixing with hydrocarbon, respectively (*Maneerat & Dikit, 2007*). Microorganisms can have high or low surface hydrophobicity depending on how they take in hydrocarbons. Microbes that can directly absorb hydrocarbons typically have highly hydrophobic surfaces. Production of cell-bound biosurfactants is connected to hydrocarbon absorption. This phenomenon is discussed in detail by *Banat et al. (2010)* who worked on Gordonia (*Bouchez-Naïtali et al., 1999*). However, when BS/BE are released extracellularly, the hydrocarbon absorption is mediated by the BS and bacteria exhibit low surface hydrophobicity (*Satpute et al., 2010*).

### Modified drop collapse method

Pennzoil is sparsely coated on microtitre plates. In the center of the well, five mL of culture broth is added, and observations are made for 1 min. The presence of biosurfactant in the culture broth is indicated if a drop of a sample collapses from the coated oil. However, a tiny amount of surfactant in the sample could result in false negative results (*Satpute et al., 2008*). On agar plates, isolates are cultivated for 24 h. At one end of the glass slide, a droplet of 0.9% NaCl is combined with a sample of the colony. The droplet is seen while the slide is tilted. By observing droplets collapsing, BS producers can be found. This method essentially modifies the drop collapse method (*Satpute et al., 2010*).

### Oil spread method

In a Petri plate, 20 mL of crude oil is added to 50 mL of distilled water (DW), and 10 mL of culture broth is added to the oil-coated water's surface. A colony encircled by an emulsified halo is seen as proof that BS is being produced (*Morikawa, Hirata & Imanaka, 2000*). It is one of the most effective ways to find biosurfactant producers (*Satpute et al., 2010*).

### Emulsification index

The emulsification index (EI) is a calculation used to gauge emulsification activity. Culture broth is given a 1:2 v/v addition of kerosene, vortexed for 2 min, and then left to stand for

24 h. The layer created between the aqueous and kerosene layers is used to determine the height of the emulsion. Some authors have reported making adjustments, such as utilizing one mL of broth, four mL of water, and six mL of kerosene vortexed to achieve the greatest amount of emulsification. EI is determined by measuring the emulsion height (*Ellaiah et al., 2002*). The EI stability denotes a surfactant's strength produced by BS manufacturers (*Haba et al., 2000*).

## Colorimetric methods

Anthrone, phenol-sulfuric acid, and orcinol are among the colorimetric techniques that are frequently utilized (*Raza et al., 2007*). In these procedures, rhamnolipids are extracted from the aqueous sample into an organic solvent, and the rhamnose and lipid portions of the molecule are then separated by acid hydrolysis. After the sugar has reacted with phenol, orcinol, or anthrone, it is then measured spectrophotometrically. These techniques are widely used for rhamnolipid quantitation, but because of the powerful acid required, they are time-consuming, difficult, and dangerous to use (*Zhu & Zhang, 2008*). They are also indirect as rhamnose is detected rather than rhamnolipids, and further steps must be taken to establish the ratio of the sugar to lipid moieties in the combination (which may alter throughout manufacture) to accurately determine rhamnolipid concentration. Additionally, several solvents, inorganic salts (such as NaCl), carbonyl or oxidizing substances, and proteins may also affect the measurement (*Heyd et al., 2008*; *Pinzon-Gamez, 2009*). The effectiveness of strain selection is increased by a quick and simple method for rhamnolipid analysis, development of processes and metabolic engineering for increased output. Subsequently, *Siegmund & Wagner (1991)* used agar plates with methylene blue and cetyl trimethyl ammonium bromide (CTAB) (SW plates), and they suggested a semi-quantitative technique. The technique depends on cationic chemicals like methylene blue or CTAB and the ability of anionic surfactants to produce insoluble ion pairs. Rhamnolipids, an anionic biosurfactant, can interact with these cationic molecules to form a complex, which can be seen by the development of dark blue patches on the agar plates (*Pinzon-Gamez, 2009*). Therefore, colonies established around rhamnolipid-producing strains can be identified by the dark blue halo, and the regions of the halo could be connected with the quantification of rhamnolipids produced. This technique is applied for strain selection and screening research which is unique to glycolipids (*Pinzon-Gamez, 2009*), especially for distinguishing between rhamnolipid producers and non-producers. This approach is frequently insufficiently quantitative to identify the strain that produces the most among many others, particularly when many of them generate other pigments that obstruct the creation or detection of the blue halo. Additionally, variables like various bacterial cell growths, cultivation times, and agar plate filling levels can all readily affect the halo diameter (*Pinzon-Gamez, 2009*).

## TOWARDS COMMERCIAL PRODUCTION OF RHAMNOLIPID

Similar to most biotechnology processes, the primary bottleneck in the manufacture of biosurfactants is the production economy. The raw materials used have a significant impact on the cost of manufacturing; in most biotechnological processes, it is estimated that raw

materials make up between 10% and 30% of the entire cost of production (*Mukherjee, Das & Sen, 2006*; *Makkar, Cameotra & Banat, 2011*). According to *Müller et al. (2010)*, the use of expensive substrates, constrained product concentrations, low yields, and the creation of product mixes rather than pure compounds are the causes of the limited usage of microbial surfactants in the industry. The usage of anti-foaming chemicals is one of many growth and upscale issues that contribute to the high costs of downstream processing (*Makkar, Cameotra & Banat, 2011*).

Therefore, it is preferable to employ inexpensive raw materials in order to lower this cost. The utilization of inexpensive, agro-based raw materials as substrates for the synthesis of biosurfactants is one option that was thoroughly investigated. The manufacturing of biosurfactants can be supported by a number of inexpensive raw materials, including plant-derived oils, oil wastes, starchy materials, lactic whey, and distillery wastes (*Mukherjee, Das & Sen, 2006*; *Saharan, Sahu & Sharma, 2011*). Although it might seem straightforward, the key issue with this strategy is choosing the appropriate waste material with the proper ratio of nutrients to support cell development and product buildup. The impact of the components on the characteristics of the finished product is another issue with this method (*Makkar, Cameotra & Banat, 2011*).

## Vegetable oils and oil wastes several studies

Rapeseed oil, babassu oil, and maize oil are examples of plant-derived oils that have demonstrated their ability to serve as efficient and affordable raw materials for the manufacturing of biosurfactants. Similarly, different bacteria produced rhamnolipid, sophorolipid, and mannosylerythritol lipid biosurfactants using vegetable oils such as sunflower and soybean oils (*Mukherjee, Das & Sen, 2006*). Growing *P. aeruginosa* 47T2 c on olive oil mill effluent (OOME) as the only carbon source (a significant waste issue in Spain) will provide rhamnolipids (*Mercad et al., 1993*), in addition to different lipophilic wastes that might be used in rhamnolipids production (*Makkar, Cameotra & Banat, 2011*).

Canola oil, soybean oil, and glucose were used as a combination for *P. aeruginosa* UW-1 to produce rhamnolipids, and the results showed a 10–12 times increase in rhamnolipid production in vegetable oils compared to glucose (*Sim, Ward & Li, 1997*). Additionally, *P. aeruginosa* UG2 cultures cultured on maize oil as the only carbon source were reported to produce a variety of rhamnoli (*Mata-Sandoval, Karns & Torrents, 2000*). *P. aeruginosa* isolate Bs20 cultured on a medium containing soybean oil also produced rhamnolipid (*Abdel-Mawgoud, Aboulwafa & Hassouna, 2009*). The rhamnolipid was generated as a thick, sticky, oily, yellowish-brown liquid with a fruity smell. It also showed very high surface activity, emulsifying ability, and qualities that allowed it to withstand heat and light. These features suggested that rhamnolipids are suitable for application in the bioremediation of hydrocarbon-contaminated locations or the petroleum industry, and this conclusion was later verified (*Perfumo et al., 2010*). Additionally, *Bacillus velezensis* S2 is expected to play a significant role in oil remediation from the environment as well as serve as a potential source of non-toxic and eco-friendly biosurfactants for various industrial applications (*Sultana et al., 2024*).

## Mixed substrates of vegetable industries

Some researchers adopted a mixed substrate strategy to make processes more cost-effective (*Haba et al., 2000*). To create stable emulsions with kerosene oil, nine *Pseudomonas* and two *Bacillus* strains were able to reduce the surface tension (to about 32–36 mN/m). *P. aeruginosa* 47T2 used waste frying cooking oil (sunflower and olive oil) as substrates and generated 2.7 g/l of rhamnolipid with a production yield of 0.34 g/g (*Makkar, Cameotra & Banat, 2011*). Using low-cost raw materials, researchers were able to produce larger quantities of biosurfactants, with yields of 4.31, 2.98, and 1.77 g/l of rhamnolipid biosurfactants using, respectively, soybean oil, safflower oil, and glycerol. DS10-129 *P. aeruginosa* (*Rahman et al., 2002*). Rapeseed oil was used as the substrate in a biotechnological process that produced 45 g/l of combinations of mono and di-rhamnolipids by *Pseudomonas* sp. DSM 2874 (*Trummler, Effenberger & Syldatk, 2003*; *Makkar, Cameotra & Banat, 2011*). The formation of rhamnolipid (1–4), L-(+) rhamnose, and (R, R)-3- (3-hydroxydecanoyloxy) decanoic acid was caused by the direct addition of Naringinase to resting cells. This was one of the first attempts to produce pure rhamnolipid using an integrated microbial/enzymatic approach (*Makkar, Cameotra & Banat, 2011*).

Another Brazilian research team isolated *P. aeruginosa* LBM10 from a southern coastal region of the country. This strain was able to grow on various inexpensive carbon sources, including fish oil, soybean oil, soybean oil soap stock, and glycerol, producing a biosurfactant of the rhamnolipid type (*Prieto et al., 2008*).

## Restaurant frying oil wastes

Remaining cooking or frying oil is a further raw material utilized in the vegetable sector. This large source of inexpensive fermentative waste is rich in nutrients. The globe over, a lot of cooking oil is produced in restaurants. According to estimates, the United States alone produces 100 billion L of oil waste per week (*Shah, Jurjevic & Badia, 2007*). Few studies have taken advantage of these frying oils' enormous potential for the generation of biosurfactants (*Zhu et al., 2007*). *P. aeruginosa* zju.u1 M. was used to produce 20 g/l rhamnolipid in a 50 L bioreactor (*De Lima et al., 2009*). Using various waste-frying soybean oils, the effectiveness and scope of biosurfactant production by the *P. aeruginosa* PACL strain were examined; a maximum rhamnose concentration of 3.3 g/l, an emulsification index of 100%, and a minimum surface tension of 26.0 mN/m was attained (*Makkar, Cameotra & Banat, 2011*).

## Lactic whey and sugar industry wastes

Lactic whey from dairy products makes a cheap and practical substrate for the manufacture of biosurfactants, which can represent a good substrate for commercial biosurfactant synthesis than synthetic media (*Mukherjee, Das & Sen, 2006*). However, research showed that molasses, with high sugar content, is a promising substrate for the generation of biosurfactants. *P. aeruginosa* GS3 produces rhamnolipid biosurfactants while growing on molasses and maize steep liquor as the main carbon and nitrogen sources (*Patel & Desai, 1997*). The product might be used in oil recovery due to strong surface activity and emulsification properties. Producing rhamnolipids from soy molasses is a resource that is readily available and reasonably priced (*Rashedi et al., 2005*), such as growing *P. aeruginosa*

EBN-8 on molasses as the only source of energy and carbon to produce a microbial surfactant. Maximum rhamnolipid yields (1.45 g/l) were attained (*Makkar, Cameotra & Banat, 2011*).

### Other unconventional substrate sources

*P. aeruginosa* MTCC 2297 produced rhamnolipids exclusively from orange fruit peeling (*George & Jayachandran, 2009*). Fewer yields were recorded when glycerol was used by *P. aeruginosa* as the only carbon source for the synthesis of rhamnolipid as opposed to conventional hydrophobic low-cost substrates (*Makkar, Cameotra & Banat, 2011*). Glycerol was used as a substrate for *P. aeruginosa* DS10-129 to produce rhamnolipid biosurfactants with a yield of 1.77 g/L (*Rahman et al., 2002*), while *P. aeruginosa* cultured on a basal mineral medium with glycerol as the only carbon source produced 15.4 g/L rhamnolipids (*Guo-Liang et al., 2005*). Using glycerol as a carbon source for microbial growth and biosurfactant synthesis is feasible (*Lee et al., 2004*). The generation of biosurfactants using fish oil for improved culture medium for *P. aeruginosa* BYK-2 KCTC 18012P resulted in an increased rhamnolipid yield of 17 g/l. The utilization of fish oil as a substrate showed good potential for the production of biosurfactants as inexpensive substrate that will aid in the cost-effective manufacturing (*Makkar, Cameotra & Banat, 2011*).

## RHAMNOLIPID APPLICATIONS

Rhamnolipids can be made from renewable resources, biodegradable, and not hazardous to people. Due to these characteristics, these glycolipids are appealing for more environmentally benign applications of surfactants (*Pinzon-Gamez, 2009*). Growth of 'green chemistry' in industry in response to the need to lessen or eliminate harmful effects on the environment and public health resulting from the overuse of chemical compounds (*Lavanya, 2024*). RLs are essential biotechnology products that are used in both industrial and medical settings. They can be utilized in the petrochemical, petroleum, environmental management, agrochemical, food and beverage, cosmetics, pharmaceutical, and mining and metallurgical industries as emulsifiers, de-emulsifiers, wetting and foaming agents, functional food components, and detergents (*Mukherjee, Das & Sen, 2006*; *Salihu, Abdulkadir & Almustapha, 2009*; *Nayak, Vijaykumar & Karegoudar, 2009*).

RLs biosurfactants are helpful as antibacterial agents and immunomodulatory compounds for medical purposes (*Muthusamy et al., 2008*; *Bhadoriya & Madoriya, 2013*). Antibacterial, antifungal, antiviral, and anticancer activities are present in several surfactants (*Ibrahim & Banat, 2019*). Biosurfactants also serve as therapeutic and probiotic agents, operate as anti-adhesive agents to infections, and are effective in the treatment of numerous diseases. They serve as an anti-adhesive and an inhibitor of the development of fibrin clots against a variety of pathogenic microorganisms (*Mulligan, 2005*; *Gudiña, Teixeira & Rodrigues, 2010*). By enhancing the apparent solubility of petroleum components and successfully lowering the interfacial tensions between oil and water *in situ*, surfactants also play a significant role in enhanced oil recovery (*Ward, Singh & Van Hamme, 2003*). Surfactants' impact on bioremediation, however, is unpredictable. Depending on the

chemical properties of the surfactant, pollutant, and microbial physiology, the addition of a surfactant of chemical or biological origin speeds up or occasionally slows down the bioremediation of contaminants (*Mulligan, 2009*).

In nature, biosurfactants are involved in stimulating the swarming movement of microorganisms, contributing to cellular physiological processes of signaling and differentiation, and enhancing the bioavailability of hydrophobic compounds (*Kearns & Losick, 2003*).

## Biosurfactant mechanism

When a single amphipathic biosurfactant molecule is added to liquid water, the hydrophobic sections of the molecule will orient above the water surface, minimizing energy interactions with polar water molecules. Naturally, polar aqueous environments will not be preferred by hydrophilic regions over the somewhat hydrophobic gas phase above the surface. This partitioning will continue until the liquid surface is saturated as additional biosurfactant molecules are added to the mixture. These additions will break the hydrogen bonds that are in charge of the water's high surface tension, resulting in a drop in surface tension. At the point of surface saturation, further surface tension reductions will be restricted, and as biosurfactant is added, molecules will be driven into the aquatic milieu. Once there are multiple biosurfactant molecules in the aqueous environment, micelle formation, in which hydrophobic biosurfactant areas cluster together and are shielded from water interactions by a hydrophilic shell, will result in the lowest energy state between the water and biosurfactant molecules. Micelles are constantly forming, dispersing, and reforming by enlisting fresh biosurfactant molecules (*Singh, Kuhad & Ward, 2009*).

## Biosurfactants *versus* synthetic surfactants

Concern for environmental preservation has boosted interest in the creation of and characteristics of biosurfactants since the last decade of the previous century (*Cohen & Exerowa, 2007*). Aside from their potential for cost-effective manufacture, biosurfactants also have the distinct advantage of being biodegradable, which makes them "green" chemicals that are good for the environment (*Whang et al., 2008*; *Salihu, Abdulkadir & Almustapha, 2009*; *Abdel-Mawgoud, Lépine & Déziel, 2010*). Even under settings of high salinity, temperature, and pH, biosurfactants can maintain their characteristics (*Ron & Rosenberg, 2002*; *Salihu, Abdulkadir & Almustapha, 2009*), and are compatible with human skin, minimal irritant (*Pornsunthorntawee et al., 2008*). Among the special qualities of these agents are their ability to increase the bioavailability of poorly soluble organic compounds, like polycyclic aromatics, as well as other significant benefits like bioavailability, activity under a variety of conditions, ecological acceptability, low toxicity, and their capacity to be modified by biotechnology and genetic engineering (*Tugrul & Cansunar, 2005*; *Salihu, Abdulkadir & Almustapha, 2009*). Therefore, using them could provide some answers for the bioremediation of polluted soil and subsurface habitats (*Salihu, Abdulkadir & Almustapha, 2009*; *Lai et al., 2009*).

Water's surface tension can be reduced by a good surfactant from 72 to 35 mN/m, while the interfacial tension between water and hexadecane can be reduced from 40 to

1 mN/m (*Mulligan, 2005*). *B. subtilis* surfactin may lower the interfacial tension of water and hexadecane to less than 1 mN/m and the surface tension of water to 25 mN/m. The surface tension of water is reduced by rhamnolipids from *P. aeruginosa* to 26 mN/m, while the interfacial tension of water and hexadecane is reduced to 1 mN/m (*Muthusamy et al., 2008*). The surface tension and interfacial tension of *T. bombicola* have been reported to be reduced to 33 mN/m and 5 mN/m, respectively (*Cavalero & Cooper, 2003*; *Muthusamy et al., 2008*). In general, biosurfactants are more effective and efficient than chemical surfactants, and their CMC is 10–40 times lower, meaning that a smaller amount of surfactant is required to achieve the greatest reduction in surface tension. The surface activities of many biosurfactants are not impacted by environmental factors like pH and temperature. Temperature (up to 50 °C), pH (4.5−9.0), and NaCl and Ca concentrations up to 50 and 25 g/l, respectively, did not affect the lichenysin of *B. licheniformis* JF-2 (*Muthusamy et al., 2008*). The surface activity of a lipopeptide from *B. subtilis* LB5a remained constant from pH 5 to 11 and up to 20% NaCl concentrations after autoclaving (121 °C/20 min) and six months at −18 °C.

Contrary to manufactured surfactants, molecules made by microbes are quickly destroyed (*Mohan, Nakhla & Yanful, 2006*; *Muthusamy et al., 2008*), especially those appropriate for use in environmental applications like bioremediation (*Mulligan, 2005*), and the cleanup of oil spills. The literature contains very little information on the toxicity of microbial surfactants. They are typically regarded as low- or non-toxic products, making them suitable for usage in food, cosmetics, and pharmaceuticals. According to one report, a synthetic anionic surfactant called Corexit had an $LC_{50}$ (concentration fatal to 50% of test species) against the bacteria *Photobacterium phosphoreum* that was ten times lower than that of rhamnolipids, indicating that the chemical-derived surfactant was more hazardous. Regarding toxicity and mutagenesis potential, a biosurfactant from *P. aeruginosa* was compared to a synthetic surfactant (Marlon A-350) that is often used in the industry. The biosurfactant was shown to be just marginally less hazardous and non-mutagenic than the chemical-derived surfactant, according to both assays (*Muthusamy et al., 2008*).

Emulsions that are stable and have a long shelf life can be created. Biosurfactants can emulsify or de-emulsify an emulsion. In general, high-molecular-mass biosurfactants outperform low-molecular-mass biosurfactants as emulsifiers. *T. bombicola* sophorolipids have been found to lower surface and interfacial tension, however they are poor emulsifiers (*Cavalero & Cooper, 2003*). Liposan, in contrast, has been successfully utilized to emulsify edible oils even though it does not diminish surface tension (*Muthusamy et al., 2008*). The fact that polymeric surfactants cover oil droplets to create stable emulsions gives them additional benefits. Making oil/water emulsions for food and cosmetics is made easier thanks to this feature. Rhamnolipids are now utilized in housekeeping and cleaning products as surface-active agents. According to OECD-tests 301F, 201, and 202, these items have little influence on aquatic life, are entirely biodegradable, and are environmentally benign (*Leitermann et al., 2010*).

## Bioremediation and enhanced oil recovery

Alkanes, cycloalkanes, aromatics, polycyclic aromatic hydrocarbons, asphaltenes, and resins are some of the many hydrocarbons found in petroleum. The best surfactants for breaking down industrially produced oil-in-water and oil-in-oil emulsions, cleaning up oil spills, and emulsifying oil (*Wang & Mulligan, 2004*; *Pinzon-Gamez, 2009*). In addition to hexadecane, tetradecane, virgin, creosote, and hydrocarbon mixes in soils, rhamnolipid addition demonstrated to promote the biodegradation of hexadecane, octadecane, n-paraffin, and phenanthrene in liquid systems (*Maier & Soberón-Chávez, 2000*). Both improved substrate solubility for the microbial cells and interaction with the cell surface, which increases the surface's hydrophobicity and makes it easier for hydrophobic substrates to associate, are potential pathways for accelerated biodegradation. The mineralization of octadecane increased to 20% from 5% for the controls at a concentration of 300 mg/L rhamnolipids (*Zhang & Miller, 1992*; *Wang & Mulligan, 2004*). During the development of hexadecane, the biosurfactant-producing strain produced more of an increase in cell surface hydrophobicity than a non-producing strain did (*Beal & Betts, 2000*). Hexadecane's solubility was elevated by rhamnolipids from 1.8 to 22.8 mg/L. There are signs that inhibition can also happen. Rhamnolipids and the fertilizer Inipol EAp-22, according to other investigations, improved the biodegradation of aromatic and aliphatic chemicals in the aqueous phase and soil reactors (*Churchill et al., 1995*). When cleaning up Exxon Valdez oil spills at Alaskan gravel, rhamnolipids and pressurized water were compared. Rhamnolipids were discovered to greatly outperform water alone in successfully releasing oil from polluted gravel (two to three times *Pinzon-Gamez, 2009*). Rhamnolipids have been examined for their ability to biodegrade as well as for their efficacy in cleaning soils tainted with petrol and other hydrocarbons in numerous research articles. An aqueous rhamnolipid solution was shown by *Neto et al. (2009)* to be 3.2 times more effective than water at removing organic matter from the soil. Rhamnolipids and various bacterial consortia have been used to demonstrate the bioremediation of n-alkanes in petroleum sludge: C8–C11 alkanes were destroyed, and alkanes in the range of C12–C21 were degraded up to 83–98% (*Pinzon-Gamez, 2009*). Primary pumping techniques only retrieve about 30% of the crude oil from the field in the petroleum industry. During water, steam, or fire flooding recovery operations, the injection of surfactants lowers the surface and interfacial tensions of the oil in the reservoir, enabling oil flow and penetration through reservoir pores (*Norman et al., 2004*; *Pinzon-Gamez, 2009*). The many processes involved in the deterioration of organic substrates are stimulated by rhamnolipids. Depending on how the substrate is given, the biodegradation process' effectiveness and the precise mechanism of action of rhamnolipid may change. This group demonstrated that as opposed to matrices with smaller pore sizes or in sea sand, matrices with pore sizes higher than 300 nm accelerated the breakdown of hexadecane to a greater extent (*Calvo et al., 2009*).

## Remediation of a heavy metal-contaminated soil by rhamnolipid

Arsenic and heavy metal exposure is linked globally to negative health impacts, according to reports (*Wang & Mulligan, 2004*; *Rodrigues et al., 2006*). Natural processes like weathering and erosion of enrichments as well as human activities like mining and smelting operations

may release these harmful elements into the environment. For polluted soils or solid wastes, several remediation strategies are proposed, including excavation and landfilling, thermal treatment, stabilization, and solidification (*Wang & Mulligan, 2004*; *Wang & Mulligan, 2009*). Due to the prohibitive costs or land space requirements, however, the adoption of these methods has been constrained (*Wang & Mulligan, 2009*). They can be combined with other substances, including salts like sodium chloride and/or additions like alcohol. By reducing interfacial tension and stimulating the formation of aqueous micelles, these chemicals would be most efficient in encouraging the mobilization of organic molecules with high lipid and relatively low water solubility. Ion exchange, precipitation-dissolution, and counter-ion binding are three potential processes for the extraction of heavy metals by surfactants. To limit the mobility of the pollutants, polymers or foams might also be added. Heavy metal-contaminated soil can be successfully treated with rhamnolipids (*Wang & Mulligan, 2009*). According to several studies, rhamnolipids work better than tap water and other synthetic surfactants in removing of metals, including cadmium, nickel, lead, and chromium. The stability constants of rhamnolipid-metal complexes are in the following order: $Cu^{2+} > Pb^{2+} > Zn^{2+} > Fe^{3+}$. Metal mobilization from sediment and soil requires rhamnolipid adsorption (*Mulligan et al., 1999*; *Ochoa-Loza, Artiola & Maier, 2001*; *Wang et al., 2008*; *Wang & Mulligan, 2009*). The negatively charged rhamnolipids may compete with arsenic for adsorption sites, squelching it and mobilizing it as a result. Arsenic mobilization by rhamnolipids may be explained by an anion exchange between the arsenic anions and the rhamnolipids (*Wang & Mulligan, 2009*).

## Application in cosmetics and personal care products

Rhamnolipids are utilized in a variety of formulations in health care products, including toothpaste, antacids, acne pads, anti-dandruff treatments, contact lens solutions, deodorants, and nail care items (*Kanlayavattanakul & Lourith, 2010*). Surfactants with high surface and emulsifying activity are necessary for these products (*Vasileva-Tonkova et al., 2001*), especially emulsifying activity, which is essential for these products' textural consistency (*Haba et al., 2003*). Due to their skin compatibility and extremely minimal skin irritation, cosmetics containing rhamnolipids have been trademarked and utilized as anti-wrinkle and anti-aging products. As a result, one business launched these products in a variety of dosage forms (Aurora Advanced Beauty Labs, Inc., St. Petersburg, FL, USA) and announced a variety of rhamnolipid-containing cosmetic items that would be sold (*Desanto, 2008*). Although, in the skincare industry, sophorolipid and rhamnolipid biosurfactants have demonstrated the potential to offer a natural, sustainable, and skin-compatible alternative to synthetically compounds, there are three main barriers for market adoption of glycolipid technology and utilization in academic research and skincare applications include: (1) low rhamnolipid yield, (2) impure preparations and/or poorly characterized congeners, (3) lack of the safety/bioactivity assessment of sophorolipids and potential pathogenicity of some native glycolipid-producing microorganisms. Consequently, more rhamnolipids testing and experimental techniques/methodologies are recommended for increasing the acceptance of glycolipid biosurfactants for use in skincare applications (*Adu et al., 2023*).

## Applications in pharmaceuticals

RLs have a variety of antibacterial characteristics (*Itoh et al., 1971*). In particular, they are proven to have an inhibition effect against a wide range of bacteria, including both Gram-negative and Gram-positive species, such as *Bacillus subtilis. Bacillus cereus* and *Rhodococcus erythropolis*. The cell envelope is the target of synthetic surfactants. Similarly, the biological membrane is intercalated into and destroyed by RLs' permeabilizing effect in their suggested mechanism of action (*Sotirova et al., 2008*). Three different strains of *P. aeruginosa* cultivated on various types of vegetable oil waste have been the subject of extensive research by the Manresa group on the antibacterial effects of mixes of RL congeners (*Benincasa, 2007*; *Abdel-Mawgoud, Lépine & Déziel, 2010*). Surfactin, rhamnolipids, and mannosileritritol lipids show functional properties including; antimicrobial, biodegradability, demulsifying and emulsifying capacity (*Begum, Saha & Mandal, 2023*; *De Oliveira Schmidt et al., 2023*). Various RL combinations showed significant antimicrobial activity against several Gram-negative species, with *Serratia marcescens*, *Enterobacter aerogenes*, and *Klebsiella pneumoniae* being the most susceptible ones, and nearly all tested Gram-positive species, including *Staphylococcus, Mycobacterium,* and *Bacillus*. They also discovered strong inhibitory efficacy against a variety of fungal species, including the phytopathogens *Botrytis cinerea* and *Rhizoctonia solanii*, the filamentous fungus *Chaetomium globosum*, *Aureobacidium pullulans*, and *Gliocladium virens*, but no discernible impact on yeasts (*Abalos et al., 2001*). Investigations were made into the algicidal potential of the rhamnolipid biosurfactants (a combination of Rha-Rha-C10-C10 and Rha-C10C10) generated by *Pseudomonas aeruginosa*. According to the findings, *Heterosigma akashiwo*, a species of harmful algal bloom (HAB), could be affected by the biosurfactants' algicidal properties. In addition, the rhamnolipids showed substantial lytic activity towards *H. akashiwo* at higher concentrations (4.0 mg/L), which significantly hindered the development of *H. akashiwo* in a medium containing them (0.4–3.0 mg/L). Rhamnolipids' effects on the growth of *Gymnodinium* sp. and *Prorocentrum dentatum*, two more HAB species, were also investigated. The cells of *P. dentatum* were inhibited or lysed at higher concentrations (1.0–10.0 mg/L) compared to the significant algicidal action on *H. akashiwo*, while the cells of *Gymnodinium* sp., were not suppressed by the same therapy, demonstrating the ability of rhamnolipids to control HABs only (*Wang, 2004*). The extent of the alga's ultrastructural damage was severe at high rhamnolipid concentrations and for long periods of contact, according to morphometric examination at the ultrastructural level using transmission electron micrographs. The plasma membrane experienced the initial reaction, which led to some disintegration. The absence of a membrane made it easier for rhamnolipid biosurfactants to enter the cells and allowed other organelles to be damaged. This led to the injury of chloroplast, vacuolization of mitochondria, deformation of the cristae, disruption of the nuclear membrane, and condensation of chromatin in the nucleus, suggesting that the lytic activity of rhamnolipids was primarily caused by their strong surface activity and their propensity to cohere on the surface afterward, the ultrastructure suffers irreparable damage and organelles lose their functionality, causing the cells to lyse (*Wang & Mulligan, 2004*).
## Applications in agricultural and plant protection

RLs were found to have strong zoosporicidal activity against a variety of zoosporic phytopathogens, including species from the Pythium, Phytophthora, and Plasmopara genera, likely by zoospore lysis (*Abdel-Mawgoud, Lépine & Déziel, 2010*). *P. aeruginosa* strain PNA1's biocontrol ability depended on the generation of RLs since rhlA mutant was substantially less effective at preventing *Pythium* sp.-caused plant disease (*Abdel-Mawgoud, Lépine & Déziel, 2010*). The FDA has authorized the use of a biofungicide made using rhamnolipids to stop plant pathogenic fungi in fruit, vegetable, and legume crops (*Nitschke & Costa, 2007*). Additionally, these formulations are thought to be low-toxic and nonmutagenic (*Pinzon-Gamez, 2009*). ZonixTM is another rhamnolipid biofungicide (Jeneil Biosurfactant Co, Saukville, WI, USA), used to protect horticulture crops from harmful fungi and to regulate them (United States Environmental Protection Agency).

Plant diseases and pests decrease agricultural yield and production quality (*Savary et al., 2019*). Despite that chemical techniques, such as pesticides and fungicides, have assisted in lowering the impact of these plant infections, their long-term usage is impractical due to their negative impacts on the biodiversity of helpful soil microorganisms and public health (*Faloye & Alatise, 2017*). Because of their lower toxicity, higher efficacy, and stability under extreme environmental conditions, biosurfactants have recently received increased attention as an alternative eco-friendly strategy for managing plant diseases. They can be made from inexpensive substrates and have a wide range of structural variations. Most significantly, they are environmentally friendly because they decompose (*Abdel-Mawgoud, Lépine & Déziel, 2010*). Rhamnolipids offer a wide range of antibacterial and antiviral activities. They were shown to be effective against Gram-negative bacteria, such as *Xanthomonas campestris*, which causes black rot in crucifers and bacterial spot disease in pepper and tomato, and Ralstonia solanacearum, which causes bacterial wilt in many Solanaceous plants (*Kim et al., 2000*). It has also been demonstrated that rhamnolipids have antifungal properties. *Colletotrichum falcatum*, which causes red rot in sugarcane, has its spore germination reduced by *Pseudomonas aeruginosa*'s extracted rhamnolipid (*Borah et al., 2016*). It has been shown that rhamnolipid plant treatments can both prevent Colletotrichum orbiculare anthracnose in cucurbits and protect pepper plants from blight disease (*Vatsa et al., 2010*). Rhamnolipids showed to exhibit antifungal activities against the molds *Cercospora kikuchii, Fusarium verticillioides, Cladosporium cucumerinum, Colletotrichum orbiculare*, Botrytis *cinerea, Cylindrocarpon destructans, Magnaporthe grisea*, and *Phytophthora capsici* in greenhouse experiments (*Borah et al., 2016*; *Monnier et al., 2018*). Rhamnolipids were found to have lytic action against *Phytophthora* sp., *Pythium* sp., and *Plasmopara* sp. zoospores in *in vitro* tests (*Perneel et al., 2008*).

Rhamnolipids' capacity to promote membrane permeability, which results in cell rupture and lysis, may be a factor in their antibacterial activity. They interact with the surface lipids and proteins, removing them from the cell membrane and altering the osmolality and the structure of the cell wall. Important membrane functions, such as transportation and energy production, are affected by this alteration in protein conformations (*Banat et al., 2010*; *Sotirova et al., 2008*).

Rhamnolipids have been shown to have the potential to boost plant immunity to diseases that are both biotrophic and necrotrophic. To protect grapevine against *Botrytis cinerea*, rhamnolipids from *Pseudomonas aeruginosa* and *Bacillus plantarii* produce reactive oxygen species (ROS) and activate MAP kinase (*Varnier et al., 2009*). Rhamnolipid-exposed plants have stimulated the Arabidopsis defense marker genes PR1, PDF1.2, and PR4 (*Sanchez et al., 2012*). They can also activate defense genes in cells of rapeseed, wheat, and tobacco (*Varnier et al., 2009*; *Monnier et al., 2018*) if rhamnolipids enter inside the lipid bilayer of the plant membrane and are situated close to the lipid phosphate group of the phospholipid bilayers, or if they require a specialized receptor in the plasma membrane of the plant cell (*Monnier et al., 2018*).

## Applications in the food industry

Due to their non-toxic properties, biodegradability, high surface and interfacial activity, high thermal and chemical stability, and sustainability through production from renewable resources, rhamnolipids have the potential to be used as emulsifiers and stabilizers in many applications of food processing, bioremediation, and food additives (*Banat, Makkar & Cameotra, 2000*). Because they can bind protein with a high affinity and improve the solid surface's wettability, they are effective for cleaning ultrafiltration (UF) membranes of protein (*Sinumvayo & Ishimwe, 2015*).

In addition to having antifungal activity against *Candida albicans*, the biosurfactants (BSs) also have an antibacterial impact against several pathogenic bacteria, including *Neisseria gonorrhoeae, Escherichia coli, Staphylococcus saprophyticus, Enterobacter aerogenes*, and *Klebsiella pneumonia* (*Giani, Zampolli & Gennaro, 2021*). The quality of packed food is improved by rhamnolipid coatings in functional food packaging because of its antibacterial, antifungal, bacteriostatic, and physicochemical qualities.

The use of rhamnolipids enhances the quality of bakery products such as bread, hamburger buns, soft rolls, Chinese steam bread, croissants, argentine bread, schnittbotchen, cake, and sponge cake in terms of dough or batter stability, volume, shape, structure, dough texture, hardness, the width of the cut, and microbiological preservation (*Magario, 2009*). The qualities of buttercream, decorating cream, and non-dairy cream filling for croissants, Danish pastries, and other fresh or frozen fine confectionery products are also improved by using them. Rhamnolipids can be used in ice cream formulations to control consistency, delay staling, solubilize flavor oils, stabilize fats, and reduce spattering in addition to preventing separation, preventing the agglomeration of fat globules, stabilizing aerated systems, and modifying rheological properties (*Giani, Zampolli & Gennaro, 2021*).

When compared to dangerous synthetic preservatives, their antimicrobial capabilities can recommend rhamnolipid for extending shelf life either directly by preventing food contamination as a food preservative or indirectly by acting as a detergent to clean surfaces in touch with the food. They increased shelf life and reduced hemophilic spores in UHT soymilk when combined with niacin. The shelf life of salad was increased and the formation of mold was reduced by adding niacin and rhamnolipids. The combination of natamycin, nisin, and rhamnolipids in cottage cheese increases shelf life by preventing the formation

of mold and bacteria, particularly gram-positive and spore-forming bacteria (*Sinumvayo & Ishimwe, 2015*).

It is simple to hydrolyze rhamnolipids to create rhamnose sugar (*Pinzon-Gamez, 2009*). It is employed in the production of various chemical compounds and as a premium flavoring agent on the market (*Maier & Soberón-Chávez, 2000*; *Pinzon-Gamez, 2009*). Currently, the production of rhamnose involves extracting quercitrin from oak bark, naringin from citrus peels, or rutin from oak bark or other plants. This technique uses caustic ingredients and produces a significant amount of toxic waste (*Chayabutra, Wu & Ju, 2001*).

## Applications as anti-biofilm agents in industrial water systems

Biofilms are bacterial communities on the surfaces of man-made and natural structures. Biofilm formation in industrial water systems such as cooling towers results in biofouling and biocorrosion represents a major health concern and economic burden. Traditionally, they were treated with alternating doses of oxidizing and non-oxidizing biocides, chemical surfactants or combination of both. Biosurfactants were shown to be powerful anti-biofilm agents and can act as biocides and biodispersants. Long-chain rhamnolipids isolated from *Burkholderia thailandensis* inhibited biofilm formation between 50% and 90%, while a lipopeptide biosurfactant from *Bacillus amyloliquefaciens* inhibited up to 96% and 99%. Additionally, the efficacy of antibiotics can be increased up to 50% when combined with biosurfactants, Thus, co-formulation with biosurfactants represents an ecological, cost-effective, and renewable solution with diminished impact when water is released into the environment. Industrial water users can involve biosurfactants combined with other molecules, such as polymers and bio-based surfactants, to proffer novel and safe alternatives (*Jimoh et al., 2023*).

# PRODUCTION ECONOMICS AND DEVELOPMENT TO OPTIMAL PRODUCTION

For biosurfactants to be commercially viable, their prices must be at least as competitive as those of their synthetic equivalents, which are currently valued at around $2/kg (*Santos et al., 2016*). The acquisition of raw materials and biomolecule recovery techniques, both of which are required to generate biosurfactants, are estimated to account for 10% to 80% of the total manufacturing cost. The upstream (selection of microbial strains, growing medium, and sterilization) and downstream (extraction and product purification) processes are all parts of the acquisition flow of any biomolecule of commercial importance. However, downstream procedures are responsible for most of the cost of biotech products (*Jimoh & Lin, 2019*). generating the same amount of biosurfactants costs 10–12 times more than generating the same amount of synthetic surfactants, which makes it difficult to integrate these biomolecules into the surfactant and related sectors (*Lotfabad, Ebadipour & Roostaazad, 2016*). Any economic analysis of a project for the production of biological products must include the anticipated capital investment, operational costs, and profitability analysis. Due to the distinctive properties of each biomolecule, the cost of biosurfactants can range from $1/kg to well over $10,000,000.00/kg. Rhamnolipids can cost anywhere from $1.5 and $1,500 per gram on the market, depending on purity

and source. Due to their potential use in the pharmaceutical and cosmetics industries, small-batch manufacture of lipopeptide biosurfactants costs between $20 and $130/mg (*Luna et al., 2012*). The resources utilized during the fermentation and purification stages of batch manufacturing largely impact the cost of biosurfactants. One method for making biosurfactants involves using solid-state fermentation using substrates obtained from other industrial processes. Solid-state fermentation has the potential to produce biosurfactants, but much more research must be done before it can be said to be economically viable. There hasn't been much research done on this fermentation method for producing biosurfactants. The majority of existing products can only be produced using batch or fed-batch processes since the solid-state process' efficiency is constrained by things like biomass growth, heat and mass transfer reactions, and sustained nutrient feeding strategies. Therefore, more research is needed to create a bioprocess that uses inexpensive waste biomass in solid-state fermentation to continuously produce and recover biosurfactants (*Singh, Patil & Rale, 2019*). Despite the increased cost, the biosurfactant industry has been gradually expanding over the past ten years and now accounts for around 5% of the global surfactant market. The compound annual growth rate (CAGR), expresses the rate of return on investment as a percentage each year (*Markets and Markets, 2017*). A recent analysis by Global Market Insights, Inc. projects that by 2025 the revenue from the biosurfactant market will increase at a CAGR of more than 5.6%. This is due to the growing worries about human health will encourage the market for biosurfactants to grow. Additionally, a CAGR of over 16.5% is anticipated between 2020 and 2026 for the global market for rhamnolipid-based biosurfactants used in food processing (*Sarubbo et al., 2022*).

## STRATEGIES FOR FEASIBLE COMMERCIAL BIOSURFACTANT PRODUCTION

To make biosurfactants more economically viable for the surfactant business, it is vital to reduce the overall cost of their synthesis and boost their productivity and yield. Using genetically modified microorganisms, creating novel, affordable downstream processes, utilizing affordable raw materials, and optimizing culture medium to increase biosurfactant concentration are a few of them (*Mohanty et al., 2021*).

## RESEARCH NEEDS AND FUTURE PERSPECTIVES

The development of foam-free rhamnolipids production processes using *Bacillus licheniformis* EL3 was evaluated for biosurfactant production under oxygen-limiting conditions in a bioreactor, using a mineral medium containing, NaNO3 and NH4Cl. The purified biosurfactant exhibited exceptional surface active properties, with minimum surface tension values and a critical micelle concentration similar to commercial surfactant, which indicated that *B. licheniformis* EL3 is a promising candidate for the development of foam-free biosurfactant production processes at an industrial scale (*Leal, Teixeira & Gudiña, 2024*).

Resource recovery would benefit from the creation of bio-based surfactants from waste streams. To enable environmentally friendly and commercially feasible production

and recovery systems; however, breakthroughs must be made. Before it can be scaled up, further research must be done on the generation of rhamnolipid biosurfactants by different bacteria using various industrial wastes/raw materials. Several studies have employed *Pseudomonas* sp. to produce rhamnolipids (*Varjani & Upasani, 2019*). Although rhamnolipid has several potential industrial applications, its high manufacturing costs prevent it from being used widely. Many different bacteria can make biosurfactants, and each one has a distinct structure, collection of characteristics, and possible application in a range of contexts (*Dell'Anno et al., 2018*). Although recovering biosurfactants from waste streams is a promising alternative for recovering resources, there are still many challenges to be resolved. After production, (1) recovery would undoubtedly increase the overall cost of producing biosurfactants from waste streams, (2) as much of the surfactant produced from waste streams as is practical from an economic standpoint, (3) care should be taken to avoid excessive costs. Cost-effective technique that would maximize the biosurfactants recovery while minimizing costs would be challenging. There may be a number of factors that have an impact on rhamnolipid formation, all of which need to be looked into in future research. The effectiveness of rhamnolipid-type biosurfactants for application in cleaning up polluted areas or enhancing oil recovery needs to be further investigated. (1) The use of rhamnolipid in severe conditions needs to be studied, and (2) it may be possible to improve its quality and production yield utilizing genetic engineering, according to future research. Innovative technology and methods are needed to increase output while lowering manufacturing costs. In-depth cost-benefit analyses are required for biosurfactant recovery schemes, (3) and the waste valorization strategy depends on large-scale rhamnolipid manufacturing using wastes as raw materials (*Varjani et al., 2021*).

## CONCLUSIONS

Research on the feasibility of producing rhamnolipids from renewable sources such as waste streams is increasing. Rhamnolipids are useful in many fields, including the petroleum and gas industry, agriculture, the cosmetics and medicine sectors, food and pharmaceutical sectors. Studying the impact of operating variables on rhamnolipids production and waste recovery requires systematic investigation. Recovering biosurfactants requires a thorough cost-benefit analysis. The use of waste materials in the synthesis of rhamnolipids has the potential to reduce the overall cost of manufacturing, which becomes especially apparent when production and purification are scaled up. The method might be made more sustainable and cost-effective by utilizing garbage as a resource, which is the key selling point.

### Funding

This work was financially supported by UTM through industrial projects no. R.J130000.7646.4C646 and R.J130000.7609.4C742. The funders had no role in study

design, data collection and analysis, or decision to publish. The funders had a role in the preparation of the manuscript.

## Grant Disclosures

The following grant information was disclosed by the authors:
UTM: R.J130000.7646.4C646, R.J130000.7609.4C742.

## Competing Interests

The authors declare there are no competing interests.

## Author Contributions

- Sanaa S.A. Kabeil conceived and designed the experiments, performed the experiments, analyzed the data, authored or reviewed drafts of the article, and approved the final draft.
- Amira M.G. Darwish conceived and designed the experiments, performed the experiments, analyzed the data, prepared figures and/or tables, authored or reviewed drafts of the article, and approved the final draft.
- Soad A. Abdelgalil conceived and designed the experiments, performed the experiments, analyzed the data, prepared figures and/or tables, and approved the final draft.
- Abdelaal Shamseldin conceived and designed the experiments, performed the experiments, analyzed the data, authored or reviewed drafts of the article, and approved the final draft.
- Abdallah Salah conceived and designed the experiments, analyzed the data, prepared figures and/or tables, authored or reviewed drafts of the article, and approved the final draft.
- Heba A.I.M. Taha conceived and designed the experiments, analyzed the data, prepared figures and/or tables, authored or reviewed drafts of the article, final editing, and approved the final draft.
- Shimaa Ismael Bashir conceived and designed the experiments, performed the experiments, analyzed the data, prepared figures and/or tables, and approved the final draft.
- Elsayed E. Hafez conceived and designed the experiments, performed the experiments, analyzed the data, authored or reviewed drafts of the article, final editing, and approved the final draft.
- Hesham Ali El-Enshasy conceived and designed the experiments, analyzed the data, authored or reviewed drafts of the article, final editing, and approved the final draft.

## Data Availability

This is a literature review.

## Supplemental Information

Supplemental information for this article can be found online at http://dx.doi.org/10.7717/peerj.18981#supplemental-information.

# PeerJ

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
