# Peer review of "Rhamnolipids bio-production and miscellaneous applications towards green technologies: a literature review"

_PeerJ, doi:10.7717/peerj.18981_

## Round 0.1 · original submission · Major Revisions

All the Queries raised by reviewer 2 should be checked properly.

**Language Note:** The review process has identified that the English language must be improved. PeerJ can provide language editing services - please contact us at [email protected] for pricing (be sure to provide your manuscript number and title). Alternatively, you should make your own arrangements to improve the language quality and provide details in your response letter. – PeerJ Staff

Reviewer 1 ·

Basic reporting

1-The authors didn’t show clear viewpoint for their work. The goal; they stated in the begaining of the manuscript was not achieved.
‎2- Poor English language throughout the whole manuscript with obvious mistakes in writing the ‎scientific names of some mentioned microorganisms.

Experimental design

1- No use of PRISMA as a tool for writing a systemic review.
2- What are the criteria (if any) of inclusion and exclusion of collected data?
3- Most of references used are old ones, however, the literatures published concerning the topic in ‎consideration are not few.

Validity of the findings

Where are the references for information listed in included tables (1-4). How can the reviewer ‎double check the accuracy of the data? and how can the reader get access to the article if he needs ‎more information???

Additional comments

I recommend the authors to make their work more concise and focus on specific point regarding the topic under discussion and they will be helped by the use of PRISMA check list for writing a systematic review.

Reviewer 2 ·

Basic reporting

Major Comments:
1. The entire work needs to be thoroughly checked for grammar and usage of the language.

2. Abstract:
The abstract has to be thoroughly updated, especially the first and second sentences, as they are unclear and hard to understand now.

3. Rationale and Intended audience:
This section is not clear. The sentences are fragmented and grammatically incorrect.

4. Misinterpretation of results and wrong citations:
Throughout the article, there are a number of inaccurate citations and misinterpretations of the results. Some findings that were covered in other review papers were discussed throughout the text, and the review paper was referenced in place of the original findings. Points 7, 8, and 9 provided examples of incorrect citations. I think there will be many more instances of this kind of thing in the text in addition to the ones I've mentioned.

5. Lack of originality:
For instance, section 3.2.2's whole content is virtually identical to a review article written in 2005 by Soberon-Chavez et al. It appears that the writers haven't added any new information to this section—they have only rewritten the lines from this review.
The majority of sections 3.2.1, 3.2.3, and 3.2.4 mirrored those found in Nitschke et al. 2005. Other than the details that were given in the 2005 publication by Nitschke et al., no new information was added.
Similar incidents might be found all throughout the text.

6. Section 3, Line 145-153, Misinterpretation of results:
The authors cited Muller et al. 2010 in which the authors added 100 µL of culture to 25 L LB media and the reaction was carried out in a 30 L batch bioreactor producing 39 g/L of rhamnolipids.
This study was highly misinterpreted as you mentioned that the bacteria create up to 100 g /L of liquid culture not even rhamnolipids. The way you are writing now implies that the bacteria produced liquid culture not rhamnolipids. And nowhere in the study does it mention 100 g /L of anything, I did not know where you get this number.

7. Section 3.2.2, line 214-215- Wrong Citation:
The control of glutamine synthase by RpoN sigma factor was suggested by Totten et al. 1990 not Soberon-Chavez et al. 2005. The review paper by Soberon-Chavez et al. 2005 highlighted this finding by Totten et al. 1990.

8. Section 3.2.2, line 217-220- Wrong Citation:
The theory that P. aeruginosa, capable of denitrification, is also using NO3 − as an electron acceptor even in the presence of oxygen is not suggested by Soberon-Chavez et al. 2005, it was put forth by Manresa et al. 1991. I believe you have just cited the review paper in which this was discussed rather than finding out the original paper.
Also, P. aeruginosa is capable of denitrification not “denitrified” (as written in your paper).

9. Section 3.2.3: line 228-229-Wrong Citation
Inhibiting activity of iron at large quantity was a finding by different studies not Soberon-Chavez et al. 2005.

10. Need for reference update:
The references listed in this work, while excellent, were from earlier research because the majority of the material gathered from review papers published between 2005 and 2010.

11. Under section 8: Rhamnolipid applications-The first sub-section 8.1 is biosurfactants discussing about the properties and mechanism of biosurfactants. I don’t know why this is included under section 8, where you were supposed to discussed about the applications. Rhamnolipids ARE biosurfactants and the authors have already discussed this in detail in other sections.

12. The sub-heading under section 8 were not uniform. In some headings, you have mentioned the sector first followed by the “applications” and in some headings “applications” was written first followed by name of sector. A more uniform wording, for example either “Applications in XX” or “XX applications” would be better.

Minor Comments:
I suggested adding “while still” just before “having” in line 55 for clarity.
Line 63, please specify what you meant by “goods”. Consider using other words.
Page 101, please consider writing “isolated from” instead of “come from”.
Please specify what you meant by “As their taxonomical classification moved more apart”.
Line 188, instead of “reasons”, “purposes” would be a better fit
Line 322-323 “Rely on the rhamnolipids' abilities as a biosurfactant and involve measuring the surface tension, which changes as the rhamnolipid concentration changes”. This sentence does not make sense.

Experimental design

Please see above comments

Validity of the findings

Please see above comments

·

Basic reporting

The structure of this article is complete, the logic of the discussion is clear, the research equations are specific and correct, the research results are sufficient, and the overall level has reached publication level.

Experimental design

The research procedures are reasonable and meet scientific requirements.

Validity of the findings

The data results and analysis of this study are sufficient, and the results obtained fully support the research setting.

Additional comments

The structure of this article is complete, the logic of the discussion is clear, the research equations are specific and correct, the research results are sufficient, and the overall level has reached publication level.

---

## Round 0.2 · Major Revisions

The queries of Reviewer 1 in the previous round have also not been addressed properly. In particular, the first part till Section 8 needs more revision on the language structure and many errors. Authors, please check the queries of reviewer 4

·

Basic reporting

The article's structure is comprehensive, the logical flow of the topic is evident, the research equations are precise and accurate, the research findings are adequate, and the overall quality meets the standards for publication.

Experimental design

The research processes are rational and adhere to scientific criteria.

Validity of the findings

The data results and analysis of this study are adequate, and they strongly corroborate the research setting.

Additional comments

The article's structure is comprehensive, the logical flow of the topic is evident, the research equations are precise and accurate, the research findings are adequate, and the overall quality meets the standards for publication.

Reviewer 5 ·

Basic reporting

The authors provide an extensive review regarding the production of rhamnolipid biosurfactants with focus on using native hosts, complemented with a collection of some simple methods utilized to detect or quantify rhamnolipid production. They provide an extensive overview over possible applications scenarios.
Language: In particular the first section (chapter 1-7) is sometimes hard to follow because of a considerable number of language error, e.g. use of ";" where ";" was meant to be used.

The research field of biosurfactants with rhamnolipid as a key case study is reviewed quite frequently e.g. PubMed IDs 38595700, 38493851, 38362900, 38312346, 38284602, 37330152, 37781531, 37493824, 37094547,37298939 in the last 12 months alone + a book with several chapters dealing with rhamnolipids (https://www.sciencedirect.com/book/9780323916974/biosurfactants) . It is not laid out, how the present review distinguishes from them, in particular considering its focus on the basic studies that were published >10 years ago.

The introduction is feasible; however I was wondering why a general introduction on biosurfactants is provided late in the manuscript (l236ff and table 4). If the authors would like to include the information in this paragraph, it might fit better into the introduction.

Experimental design

The search methodology apparently overlooked studies on recombiannt production of rhamnolipids. Of course it is feasable to just focus an P. aeruginosa-Based production but this should be than pointed out explicitly, as recombinantly produced rhamnolipids marketed by the European company Evonik constitute the most prominent example of commercial production of rhamnolipids at large scale .

As mentioned, the structure is sometimes confusing with the general biosurfactant introduction in the middle of the manuscript or sudden and explained changes out of the rhamnolipid focus to unrelated organism and compounds where the relation the review's topic is not made clear:
L 126-13: "randomly" introduced Lactobacillus chapter
l. 146: Referring to Candida antarctica process . Are there no studies on the influence of pH on P. aeruginosa around?
l.184-187: the statement of properties of Surfaction, MELs and RLs appears to be micplaced in the micronutrients paragraph
l. 236-262 Description of biosurfactants in general.

Validity of the findings

Despite missing recent developments, the goals set out were matched properly

Additional comments

Specific comments:

Formalitiies:
"beta-hydroxy decanoic acids" or "β-hydroxy decanoic acids" should be used, not "b-hydroxy decaonic acids" (e.g. l. 71)
the unit of titers should read mg L -1 (as superscript) or mg/L. mg L1 does not make any sense.
l. 217 what is meant with "-L-rhamnopyranosyl-(1,2)"
l. 218-220. This sentence does not make sence. Rhamnopyranosyl-3-hydroxydecanoyl-3-hydroxydecanoate is just the chemical name for one monorhamnolipid species. The abbreviation should read Rha-C10-C10 (not Rha- Rha-C10-C10).
l.875 typo "Pseudomonas aeruginosa

Scientific comments:
l. 54: Sythesis from renewable resources is not necessarily connected to good biodegradability (although of course, both should be true for rhamnolipids (however, there are very few studies around that really characterize the biodegradation to my knowledge).

l. 103. P. putida KT2440 does not produce rhamnolipids by nature. The studies and approaches using this strain apply engineered strains expression e.g. the genes from P. aeruginosa.
As mentioned, the manuscript lacks completely a description of recombinant approaches (e.g. reviewed in doi: 10.1016/j.copbio.2020.03.007 and doi: 10.1016/B978-0-323-91697-4.00008-9) , which is fine if the authors want to focus on GMO-free production using wildtype strains. However, the statement in line 103 is misleading nonetheless.

l.122 states that "some" RL-producing bacteria were shown. How were these selected?
Why was a table from 2010 shown? it lacks recent discoveries like doi: 10.1016/j.carres.2023.108991
Liu, M. D., Du, Y., Koupaei, S. K., Kim, N. R., Fischer, M. S., Zhang, W., et al. (2024). . ISME J. doi: 10.1093/ISMEJO/WRAE022
The literature report on rhamnolipid producing organism was described to be seriously flawed. In this light, pioneers in the field suggested criteria to proof a rhamnolipid producer (doi: 10.1007/s00253-017-8262-0). I think the chapter 2 might benefit from picking this up to provide evaluation of the all the reports of different bacteria.

l. 149ff: Reports on RNA-Thermometers in the rhl-promoter should be mentioned is this context (Noll, doi: 10.1186/s13568-019-0883-5)

l235. The table 3 does not provide any information on the type of rhamnolipid. i do not know if there happened a kind of confusion;however the table 3 provided with the manuscript should be entitled with something like "conditions applied for rhamnolipid production with P. aeruginosa

Chapter 6 lacks some recent developments
doi: 10.1111/j.1462-2920.2011.02534.x
doi: 10.3389/fbioe.2020.00958
doi: 10.1007/10_2021_174
doi: 10.3389/fbioe.2023.1253652
doi: 10.1007/s00216-016-9353-y
doi: 10.1016/j.carres.2023.108991
doi: 10.1016/j.carres.2018.10.009

---

## Round 0.3 · Minor Revisions

Authors may try to address the interesting points raised by reviewer5 - they have pointed out again that there are many similar studies already. The authors need to clarify how their paper differentiates itself.

Reviewer 2 ·

Basic reporting

All comments has been addressed in the revised manuscript

Experimental design

All comments has been addressed in the revised manuscript

Validity of the findings

All comments has been addressed in the revised manuscript

Additional comments

All comments has been addressed in the revised manuscript

Reviewer 5 ·

Basic reporting

The authors made considerable efforts to improve the review. However, some comments were not addressed or even answered in the rebuttal, including

In particular, the first section is sometimes hard to follow because of a considerable number of language errors, e.g. use of ";" where "," was meant to be used.

As mentioned, the research field of biosurfactants with rhamnolipid as a key case study is reviewed quite frequently e.g. PubMed IDs 38595700, 38493851, 38362900, 38312346, 38284602, 37330152, 37781531, 37493824, 37094547,37298939 in the last 12 months alone + a book with several chapters dealing with rhamnolipids (https://www.sciencedirect.com/book/9780323916974/biosurfactants)
It would be helpful if the authors explain, e.g. in "Rationale and intended audience", how the present review distinguishes from them. The authors included now some recent studies, but I still got the feeling, that the focus of the review is on the basic studies that were published >10 years ago.

With the recombinant process of Evonik being THE example of successful rhamnolipid commercialization in the European area I still think that this approach should be mentioned even if the the authors decide to focus on non-GMO production approaches

Of course, a review can stand without addressing these points but I'll leave it to the editors to make a decision if the manuscript matches the scope.

Experimental design

Nothing more to add. Other comments were addressed properly

Validity of the findings

Nothing more to add. Other comments were addressed properly

---

## Round 0.4 · Major Revisions

The Section Editors have examined your article and have provided the following requests:

The authors should consider using recognized search engines, such as Google Scholar, WoS, and ResearchGate to verify that they have the main studies in their field.

The authors should explain why the majority of papers are so outdated.

Finally, they should focus on more recent work by seeing newly published papers, and provide a clear rationale for the importance of the area.

Reviewer 5 ·

Basic reporting

Thanks to the authors for considering the comments.

Experimental design

Nothing more to add.

Validity of the findings

Nothing more to add.

---

## Round 0.5 · accepted · Accept

The authors have addressed all of the reviewers' comments, this manuscript is ready for publication.